# Investigating the Alleviating Effects of Dihydromyricetin on Subclinical Mastitis in Dairy Cows: Insights from Gut Microbiota and Metabolomic Analysis

**DOI:** 10.3390/microorganisms13081890

**Published:** 2025-08-13

**Authors:** Jie Yu, Yingnan Ao, Hongbo Chen, Tinxian Deng, Chenhui Liu, Dingfa Wang, Pingmin Wan, Min Xiang, Lei Cheng

**Affiliations:** 1Institute of Animal Science and Veterinary Medicine, Wuhan Academy of Agricultural Sciences, Wuhan 430208, China; yujiehzau@163.com (J.Y.); ayn15849077718@163.com (Y.A.); liuchenhui@wuhanagri.com (C.L.); wangdingfa@wuhanagri.com (D.W.); wanpingmin@wuhanagri.com (P.W.); xiangmin@wuhanagri.com (M.X.); 2Hubei Provincial Center of Technology Innovation for Domestic Animal Breeding, School of Animal Science and Nutritional Engineering, Wuhan Polytechnic University, Wuhan 430023, China; chenhongbo@whpu.edu.cn (H.C.); txdeng2024@whpu.edu.cn (T.D.)

**Keywords:** dihydromyricetin, dairy cows, subclinical mastitis, fecal microbial community, metabolomics

## Abstract

Mastitis is a common disease for dairy cows that exerts tremendously detrimental impacts on the productivity of cows and economic viability of pasture. Dihydromyricetin (DMY) is a flavonoid monomeric compound that possesses anti-inflammatory and antioxidant activity. This study aimed at dissecting the effects of DMY on the lactation performance, blood parameters, gut microbiota, and metabolite profiles of dairy cows with subclinical mastitis (SM). The results showed that dietary supplementation with DMY resulted in a reduction in milk somatic cell count, an increase in serum T-AOC and CAT activity, as well as a decrease in serum MDA content. DMY significantly enhanced the prevalence of *Coprococcus* and *Roseburia* and reduced the proportion of Cyanobacteria, Proteobacteria, and *Dehalobacterium*. The amino acid degradation, antibiotic resistance, and O-antigen building blocks biosynthesis (*E. coli*) capacity of gut microbes were notably diminished by DMY supplementation in cows with SM. Moreover, fecal and plasma metabolomic analysis revealed that DMY intervention reduced the abundance of pro-inflammatory metabolites including arachidonic acid analogues, ω-6 PUFA, and structural components of bacteria. Nevertheless, the levels of anti-inflammatory and antioxidant metabolites involving secondary bile acids, antioxidant vitamins, specific amino acid analogues, etc. were elevated by DMY administration. Overall, DMY might ameliorate SM via enhancing antioxidant capacity and improving the structure of the hindgut microbial community and metabolite profiles in dairy cows. These findings underscore the potential of DMY as a valuable dietary supplement for the improvement of mammary inflammatory diseases in dairy cows.

## 1. Introduction

Mastitis is the inflammatory response of mammary glands triggered by various factors that is commonly observed in female mammals, including postpartum lactating women and dairy cows, which poses serious threats to both maternal–infant health and the economic benefits of the livestock industry [1,2]. Specifically in dairy farming, mastitis not only leads to a reduction in milk yield and deterioration of milk quality, but also increases culling rates and veterinary costs, thereby imposing substantial economic burdens on the dairy industry [2,3]. Due to the strong insidiousness, long incubation period, and high incidence of subclinical mastitis (SM), the economic losses caused by SM are much higher than those of clinical mastitis (CM) [4,5]. At present, the application of antibiotics such as ampicillin, streptomycin, and enrofloxacin is the main method for the treatment of mastitis, while the resulting bacterial resistance leads to unsatisfactory treatment effects [6,7,8]. Therefore, under the global strategy to cope with the increasingly serious problem of bacterial resistance to antibiotics, it is of great significance to explore safer and more effective interventions to modulate or control mastitis in dairy cows.

Given the fact that the complex symbiotic relationship between gut microbiota and host plays a pivotal role in regulating metabolism and immune homeostasis, a growing body of evidence has confirmed the associations between gut microbiota and inflammatory and metabolic diseases of the host [9,10]. Recent studies have pinpointed that the disturbance of gastrointestinal flora is related to the occurrence of mastitis in the host and might be an important endogenous factor contributing to mastitis [11,12,13]. A previous study showed that the disordered microbiota were present in the gut of cows with SM, mainly manifesting as a significant decrease in the abundance of *Bifidobacterium*, *Romboutsia*, *Lachnospiraceae_NK3A20_group*, *Coprococcus*, *Prevotellaceae_UCG-003*, *Ruminococcus*, and *Alistipes* alongside the substantial increase in the abundance of *Paeniclostridium* and *Klebsiella* [14]. Additionally, this dysbiosis was accompanied by a significant elevation in the levels of pro-inflammatory lipid metabolites and a notable decrease in the content of secondary bile acids (SBAs) in feces as well as the occurrence of oxidative stress [14]. Hence, the modulation of intestinal microflora and metabolites may be one of the promising means to develop interventions to alleviate mastitis in dairy cows.

Dihydromyricetin (DMY), a flavonoid monomer (C_15_H_12_O_8_, MW 320.25) belonging to the dihydroflavonol subclass, possesses a characteristic structure featuring a dihydroflavonol core with adjacent phenolic hydroxyl groups and a highly conjugated pyranone (Figure 1) [15]. DMY is the primary bioactive component from a traditional Chinese natural plant *Ampelopsis grossedentata* and is identified in grape, red bayberry, *Hesperis pendula, Astragalus angustifolius,* etc. [16,17,18,19]. The content of DMY in *Ampelopsis grossedentata* was reported to be as high as about 30–40% [16,20]. DMY is stable under acidic conditions and readily soluble in hot water and ethanol [16]. It has been proved that DMY possesses various physiological functions, including anti-inflammation, antioxidation, regulation of glucose and lipid metabolism, anti-tumor, and protection of liver function [21,22,23], making it an ideal candidate phytogenic supplement for the prevention and alleviation of mastitis in dairy cows. Dong et al. demonstrated that DMY could improve intestinal barrier function and alleviate dextran sulfate sodium (DSS)-induced colitis by restoring the balance of intestinal flora and bile acid metabolism and increasing the abundance of *Lactobacillus* and *Akkermansia* genera as well as the gastrointestinal levels of chenodeoxycholic acid and lithocholic acid in mice [24]. Dietary supplementation of DMY improved lipid metabolism and significantly increased the antioxidant capacity of serum and liver in finishing pigs [25]; moreover, it attenuated intestinal integrity impairment and inflammation caused by enterotoxigenic *Escherichia coli* K88 challenge in piglets [26]. Nevertheless, the application potential of DMY in alleviating mastitis of dairy cows has not been explored, and whether DMY could regulate mastitis by modulating the gastrointestinal flora and host metabolism in dairy cows remains indistinct and needs to be elucidated. It is well known that the efficacy of flavonoid compounds is influenced by their bioavailability. However, little information is available on the absorption profiles, metabolism, and distribution of DMY following oral administration in humans or domestic animals, especially in ruminants. The findings from studies in rats and Caco-2 cells revealed that DMY is primarily absorbed and metabolized in the gastrointestinal tract through passive diffusion and transporter-mediated mechanisms, with a portion being absorbed into the blood [21,27,28,29]. Although rumen metabolism is known to modify dietary flavonoids through processes including reduction and dehydroxylation, etc. [30,31], the resulting metabolites or rumen-bypassed prototype compounds reaching the hindgut might directly modulate the gut microbiota and its metabolic outputs or exert systemic effects after absorption. Therefore, understanding the effects of DMY via the gut microbiota–mammary axis is equally crucial for interpreting its potential modes of action in dairy cows with mastitis. This study aims to dissect the regulatory role of DMY on SM in dairy cows and investigate whether its beneficial effects are closely related to gut microbiota and metabolism. Herein, the effects of dietary DMY supplementation on lactation performance, blood parameters, gut microbiota structure, and metabolite composition were analyzed in dairy cows with SM, which contributes to clarifying the mitigation effects and regulatory mechanisms of DMY on mastitis and provide a theoretical reference for the formulation of nutritional regulation strategies for SM in dairy cows.

## 2. Materials and Methods

### 2.1. Experimental Animals and Design

This trial was conducted in the experimental dairy farm of Wuhan Academy of Agricultural Sciences. Firstly, lactating Holstein cows with similar parity (2.2), lactation days (149 ± 68 d), and milk yield (19.50 ± 6.12 kg/d) were selected. Veterinarians from Wuhan Academy of Agricultural Sciences comprehensively evaluated the udder health status of the experimental cows based on the milk somatic cell count (SCC) over the previous nearly one month and the clinical manifestations of mastitis. The widely accepted optimal milk SCC threshold for diagnosing and differentiating between healthy and mastitis-affected cows is 20 × 10^4^ cells/mL [32,33,34]. Ultimately, 20 cows with SM (SCC > 50 × 10^4^ cells/mL, no clinical manifestations of redness, swelling, and fever in the udder) were screened out and randomly divided into two groups (10 cows per group): a subclinical mastitis group (SM) and a subclinical mastitis supplemented with dihydromyricetin group (SM-DMY). These cows, housed in individual tie stalls, were fed ad libitum and had free access to water. The basal diet of this study was a total mixed ration (TMR) containing 40% roughage and 60% concentrate (Appendix A), which was offered twice daily at 5:00 a.m. and 5:00 p.m. The experimental cows in the SM group were fed a basal diet; however, cows in the SM-DMY group were fed an experimental diet supplemented with DMY (98% purity) in the basal diet. Prior to the start of the trial, an in vitro rumen fermentation simulation experiment was carried out to probe the appropriate dose range of DMY supplementation. The specific protocol for in vitro rumen fermentation simulation technique was referenced to McDougall [35]. In brief, DMY of 0, 0.05%, 0.1%, 0.15%, 0.2%, and 0.25% of dry matter (DM) were added into 0.22 g dried TMR samples (the TMR used in the in vitro trial was the same as the present study), which were incubated in a bottle with rumen fluid and freshly prepared anaerobic buffer at 39 °C for 12 h, 24 h, and 48 h. The gas production, pH, degradation rate of dry matter (DM), and content of NH3-N and short-chain fatty acids (SCFAs) were then tested. According to the results of in vitro rumen fermentation parameters, 0.05% DMY (98% purity) was added into the basal diet that was offered to the cows of SM-DMY group. The animal experiment lasted for 60 consecutive days. DMY used in this study was isolated from *Ampelopsis grossedentata* provided by Hubei Jinrui Biotechnology Co., Ltd (Enshi Tujia and Miao Autonomous Prefecture, China).

### 2.2. Milk Yield and Composition Analysis

The cows were milked two times a day at 5:30 and 17:30 by employing an automatic milking system (Afimilk, Kibbutz Afikim, Israel). The milk yield was recorded using a computerized dairy farm management system (Afimilk, Kibbutz Afikim, Israel). The average daily milk production in the last 3 consecutive days before the end of the trial was used as a representative of the milk yield during the experimental period. The milk samples were collected on day 0 before DMY supplementation and day 60 after DMY supplementation. A 50 mL sterile centrifuge tube was applied to collected milk samples from each cow. All the milk samples were dosed with potassium dichromate as preservative and stored at 4 °C and were then analyzed by using a milk composition analyzer (FT600, Foss Electric, Hillerød, Denmark).

### 2.3. Blood and Feces Specimen Collection

Blood and feces samples were collected from 7 cows randomly selected from each group on day 60 of the experimental period. The blood samples, measuring approximately 10 mL, were procured from the tail vein of cows to separate serum and plasma specimens. In addition, fresh feces samples were collected directly from the rectum of cows using sterile long-arm gloves. These samples were then divided into three aseptic 5 mL cryovials for proper preservation. All the samples were snap-frozen with liquid nitrogen and stored at −80 °C for subsequent analysis.

### 2.4. Serum Biochemical and Antioxidant Indices and Inflammatory Cytokines Measurements

A fully automatic biochemical analyzer (TBA120FR, Canon, Tokyo, Japan,) was applied to determine the concentrations of total protein (TP), albumin (ALB), globulin (GLB), uric acid (UA), triglyceride (TG), total cholesterol (TC), high-density lipoprotein cholesterol (HDL-C) and low-density lipoprotein cholesterol content (LDL-C). Alkaline phosphatase (AKP), non-esterified fatty acid (NEFA), lactate dehydrogenase (LDH), total antioxidant capacity (T-AOC), malondialdehyde (MDA), superoxide dismutase (SOD), glutathione peroxidase (GSH-PX), and catalase (CAT) were assessed following the protocol of kits purchased from Nanjing Jiancheng Bioengineering Institute. ELISA kits were used to detect TNF-α, IL-1β, IL-2, IL-8, and IL-10 levels following the protocols of the manufacturer (Shanghai Enzyme-linked Biotechnology Co., Ltd., Shanghai, China).

### 2.5. Gut Microbiota Profiling with 16S rDNA Amplicon Sequencing

MagBeads FastDNA Kit (116570384, MP Biomedicals, Santa Ana, CA, USA) was employed to isolate total genomic DNA from feces samples in accordance with the manufacturer’s instructions. The quality and quantity of extracted DNA samples were assessed by agarose gel electrophoresis and NanoDrop NC2000 spectrophotometer (Thermo Fisher Scientific, Waltham, MA, USA). The extracted DNA samples were used as templates for PCR amplification of bacterial 16S rRNA genes V3–V4 region using the primers 338F/806R. The final PCR amplicons were purified with Vazyme VAHTSTM DNA Clean Beads (Vazyme, Nanjing, China) and quantified using the Quant-iT PicoGreen dsDNA Assay Kit (Invitrogen, Carlsbad, CA, USA). Then, purified amplicons were pooled in equal amounts for pair-end 2 × 250 bp sequencing on the Illumina NovaSeq platform.

The sequence data underwent a systemic processing and analysis utilizing QIIME2 (version 2019.4), with minor modifications made in line with the guidelines provided in the official tutorials available at the QIIME2 website. In brief, the raw sequences were subjected to quality filtering to ensure reliability followed by denoising to eliminate potential errors. Subsequently, the data were merged to create a comprehensive dataset, and any chimeric sequences were removed using the DADA2 plugin. This workflow facilitated the generation of high-quality sequence data essential for accurate downstream analyses. Non-singleton amplicon sequence variants (ASVs) were aligned with mafft, and phylogeny were constructed with fasttree2. Taxonomy assignments for ASVs were conducted using the naive Bayes taxonomy classifier from the classify-sklearn tool integrated into the feature-classifier plugin, which relied on the comprehensive data provided by the Greengenes 13.8 database. Alpha diversity was analyzed to assess the complexity of species by calculating observed species, Chao1, Shannon, Simpson, Pielou’s and Good’s coverage indices with Mothur (v1.31.2). Beta diversity, presented by principal coordinate analysis (PCoA), was performed to inquire into the structural variation in microflora among samples utilizing Bray–Curtis metrics. The differences in microbiota composition across groups were evaluated for significance using permutational multivariate analysis of variance (PERMANOVA) through QIIME2. The marker taxa between groups were identified using the Linear discriminant analysis effect size (LEfSe) algorithm. Functional analysis of gut microbiota was predicted by Phylogenetic investigation of communities by reconstruction of unobserved states (PICRUSt2) [36] based on MetaCyc databases. The differences in microbial functions among groups were compared by applying the STAMP software (v2.1.3).

### 2.6. Untargeted Metabolomics Profiling

The thawed feces (50 mg) and plasma (100 µL) samples were combined with 400 μL of cold extraction solution (methanol/acetonitrile, 1:1 (*v*/*v*)), which were vortexed for 30 s and sonicated in a water bath at 4 °C for 10 min. To precipitate proteins, both the plasma and feces samples were allowed to incubate for 1 h at −40 °C, followed by centrifugating at 14,000× *g* for 20 min at 4 °C. The supernatant was dried in a vacuum centrifuge and redissolved in 100 µL acetonitrile/water (1:1, *v*/*v*) solvent. The mixtures were centrifugated at 14,000× *g* for 15 min at 4 °C to acquire the supernatant for LC-MS analysis. The quality control (QC) samples were prepared by pooling 10 μL of each feces or plasma sample, which were inserted regularly and analyzed in every 5 samples. The LC-MS analyses were performed using UHPLC (Vanquish UHPLC, Thermo Fisher Scientific, Waltham, MA, USA) coupled with a Q Exactive HFX Hybrid Quadrupole Orbitrap mass spectrometer (Thermo Fisher Scientific, Waltham, MA, USA). Chromatography separation was performed with an ACQUITY UPLC^®^ HSS T3 column (2.1 mm × 100 mm, 1.8 µm, Waters, Milford, MA, USA) at flow rate and injection volume of 0.30 mL/min and 2 μL, respectively. The column oven was maintained at 40 °C. The mobile phase consisted of 0.1% (*v*/*v*) formic acid/water (A) and 0.1% (*v*/*v*) formic acid/acetonitrile (B). Gradient elution conditions were set as follows: 0~1 min, 0% B; 1–12 min, 0 to 95% B; 12–13 min, 95% B; 13–13.1 min, 95% to 0 B; 13.1–17 min, 0% B. The primary and secondary spectra were collected by high-resolution mass spectrometry system (Q Exactive HFX, Thermo Fisher Scientific, Waltham, MA, USA). Simultaneous acquisitions using both MS1 and MS/MS (in Full MS-ddMS2 mode with data-dependent MS/MS) were utilized. The parameters were configured as follows: sheath gas pressure, 40 arb; aux gas flow, 10 arb; spray voltage 3.0 kV for ESI (+) and −2.80 kV for ESI (−); capillary temperature, 325 °C; MS1 scan *m*/*z* range, 100–1000; MS1 resolving power, 70,000 FWHM; MS/MS resolving power, 17,500 FWHM; number of data dependent scans per cycle, 10; normalized collision energy, 30 eV.

The raw MS data were acquired on a Q-Exactive using Xcalibur 4.1 and processed with Progenesis QI software (v3.0, Waters, Milford, MA, USA) including baseline filtering, peak identification, integration, retention time correction, and peak alignment. The method for correcting LOESS signals, which utilizes QC samples, was employed to adjust data between different batches and to address errors associated with instrument batches. During the quality control of data, substances with an RSD > 30% in QC samples were excluded. The identification of metabolites was carried out by comparing the precise *m*/*z* values (within 10 ppm) and MS/MS spectra against databases such as HMDB, LipidMaps, mzCloud, MassBank, and KEGG as well as a self-constructed database established with available authentic standards. Dimension reduction analyses, such as PCA, PLS-DA, and OPLS-DA, were performed on the sample datasets through “Ropls” in R package (v1.22.0) to characterize the variations in metabolite compositions across different samples. The variable importance in projection (VIP) value from the OPLS-DA model was calculated to assess its contribution to the classification. The significant differential metabolites were declared as VIP > 1, *p*-value < 0.05, and fold change (FC) > 1.2 or < 0.83. Cluster analysis of differential metabolites was performed by employing the Pheatmap package (v1.0.12) within R. KEGG enrichment analysis of differential metabolites was conducted through clusterProfiler package (v 4.6.0). The machine learning analysis of differential metabolites was undertaken by using mlr3verse package (v0.2.7).

### 2.7. Statistical Analysis

The sample size calculation of this study was performed by employing SPSS27.0 software (IBM, Chicago, IL, USA). A subsequent post hoc power analysis showed that the sample size utilized in this research achieved a statistical power of 0.85 at a significance level of 0.05 for a two-tailed test. The data were expressed as mean ± standard error of the mean (SEM). The statistical analyses were performed by GraphPad Prism 9.0 software. Unpaired two-tailed *t*-test (Student’s *t*-test) was used to compare the differences between the two groups for serum biochemical and antioxidant indices, inflammatory cytokines content, and alpha diversity index. The milk SCC and relative abundance of bacterial taxa between groups were compared by Mann–Whitney U test. Spearman correlation analysis between differential metabolites in feces and plasma and fecal microflora was performed by applying the correlation function in R package (v4.0.3) and represented by a heatmap; *p* < 0.05 was regarded as statistically significant, while 0.05 < *p* < 0.10 were interpreted as a tendency.

## 3. Results

### 3.1. Effects of Dietary DMY Supplementation on Milk Composition

As shown in Table 1, no significant differences in milk yield, milk fat, milk protein, fat-to-protein ratio (F/P), or milk urea nitrogen content were observed among the SM and SM-DMY groups (*p* > 0.05). However, milk SCCs were significantly decreased in the SM-DMY group compared with the SM group (*p* < 0.01).

### 3.2. Effects of Dietary DMY Supplementation on Blood Parameters

The measurements of blood biochemical indices showed that there were no significant differences in serum TP, ALB, GLB, AST, ALT, BUN, UA, TG, TC, LDL-C, NEFA, and LDH levels between the SM and SM-DMY groups (*p* > 0.05) (Table 2). Compared with the SM group, serum AKP (*p* = 0.086) levels tended to decrease, while HDL-C (*p* = 0.0656) content showed a tendency of increase, in the SM-DMY group (Table 2). According to the results of cytokines examination, no significant effects of dietary DMY supplementation were found for serum IL-1β, IL-2, IL-8, IL-10, or TNF-α (*p* > 0.05) (Table 3). In terms of antioxidant capacity, serum T-AOC and CAT in the SM-DMY group were notably higher than those in the SM group (*p* < 0.05), MDA content was dramatically lower than that in the SM group (*p* < 0.05), whereas there were no significant differences in SOD and GSH-PX between the two groups (*p* > 0.05) (Table 4).

### 3.3. Effects of Dietary DMY Supplementation on the Diversity and Taxonomic Composition of Gut Microbiota

To systematically dissect the regulatory mechanisms of DMY intervention on mastitis in dairy cows, 16S rDNA sequencing was performed to analyze the effects of DMY on the fecal microbiome of cows with SM. A total of 703,837 high-quality sequences were acquired from 14 fecal samples after filtering, denoising, merging, and chimera removing of 1,350,054 raw reads, with an average of 50,274 high-quality sequences per sample. Furthermore, 23,227 ASVs were obtained from each sample after normalization based on a 97% sequence similarity threshold. The dilution curves of all samples reached a stable plateau phase, indicating that the sequencing depth was adequate (Figure 2a). The microbial information detected in the current study covered the vast majority of species in the samples, which could be applied to subsequent analysis. The ASVs distribution of fecal microbiota between the two groups is shown in Figure 2b, with 4013 ASVs shared by the SM group and SM-DMY group. The results of alpha diversity analysis showed that DMY intervention had no significant effects on observed species, Chao1, Goods coverage, or Shannon and Simpson indices of fecal flora in cows with SM (*p* > 0.05) (Figure 2c–g). Beta diversity analysis based on PCoA indicated separate clusters of fecal microbiota between the SM group and SM-DMY group (Figure 2h). Moreover, Bray–Curtis-based PERMANOVA analysis revealed that the distance of fecal microbiota in the SM-DMY group was significantly greater than that in the SM group (*p* < 0.01) (Figure 2i), indicating significant differences in gut microbiota between cows in the SM group and SM-DMY group.

The relative abundance of gut microbiota was determined at different taxonomic levels to identify the changes in gut microbial profiles, which were visualized by linked bar plots. The identified ASVs were clustered in 24 phyla and 270 genera. At the phylum level, Firmicutes (64.38%, 60.97%), Bacteroidetes (30.63%, 34.59%), Tenericutes (1.77%, 1.50%), and Spirochaetes (0.73%, 1.21%) dominated in the gut of cows in the SM group and SM-DMY group, respectively (Figure 2j; Appendix A). At the genus level, *5-7N15* (3.52%, 2.97%), *CF231* (2.88%, 2.81%), *Oscillospira* (3.16%, 2.38%), *Ruminococcus* (2.07%, 2.43%), *Clostridium* (1.87%, 1.70%), *Treponema* (0.59%, 0.92%), *Paludibacter* (0.54%, 0.78%), *YRC22* (0.51%, 0.80%), *rc4-4* (0.91%, 0.37%), *Dorea* (0.58%, 0.70%), and *Coprococcus* (0.42%, 0.61%) were the most abundant genera in the SM group and SM-DMY group, respectively (Figure 2k; Appendix A).

### 3.4. Differences in Gut Microbial Composition and Function Between Groups

LEfSe analysis was devoted to identify the differential microorganisms intervened by DMY supplementation. The LEfSe cladogram plot of differential bacterial between the SM group and SM-DMY group is shown in Figure 3a. Compared with the SM group, the relative abundances of Cyanobacteria (*p* < 0.05), Proteobacteria (*p* < 0.01), Elusimicrobia (*p* < 0.05), *Phascolarctobacterium* (*p* < 0.05), and *Dehalobacterium* (*p* < 0.01) in the SM-DMY group were significantly reduced (Figure 3b). Furthermore, Bacteroidetes (*p* < 0.05), *Roseburia* (*p* < 0.05), and *Coprococcus* (*p* < 0.05) were remarkably more abundant in the SM-DMY group in comparison with the SM group (Figure 3b). Accordingly, the changes in gut microbial composition and relative abundance induced by DMY supplementation were also reflected in the functional changes in intestinal microorganisms. PICRUSt2 functional prediction revealed that the capacity of amino acid degradation (*p* < 0.0001), C1 compound utilization and assimilation (*p* < 0.0001), inorganic nutrient metabolism (*p* < 0.05), antibiotic resistance (*p* < 0.05), and O-antigen building blocks biosynthesis (*E. coli*) (*p* < 0.05) were observably declined in the SM-DMY group, while the microbes associated with pentose phosphate pathways (*p* < 0.01), secondary metabolite degradation (*p* < 0.05), methyl ketone body biosynthesis (*p* < 0.05), and secondary metabolite biosynthesis (*p* < 0.05) were notably enhanced in the SM-DMY group. Furthermore, the functions of the tricarboxylic acid cycle (*p* = 0.054), fatty acid and lipid degradation (*p* = 0.065), glycolysis (*p* = 0.072), nucleic acid processing (*p* = 0.080), and carbohydrate degradation (*p* = 0.088) tended to be improved (Figure 3c).

### 3.5. Effects of Dietary DMY Supplementation on Fecal Metabolomic Profiles

To further elucidate the impacts of DMY-regulated gut microbiota on mastitis in dairy cows, we analyzed the effects of DMY on fecal metabolomic profiles of SM cows utilizing untargeted metabolomics techniques. We detected 6885 and 4546 metabolites in all feces samples under positive and negative ion mode, respectively, among which a total of 623 named metabolites were finally identified through matching with a public and a self-established database. The results of partial least-squares discriminant analysis (PLS-DA) of each fecal sample are presented in Figure 4a,b, and they indicate that the fecal samples of the SM group and SM-DMY group can be effectively distinguished under both positive and negative ion mode. By univariate statistical analysis and orthogonal partial least-squares discriminant analysis (OPLS-DA) modeling, 58 differential metabolites were screened in the feces of cows in the SM vs. SM-DMY group, which included 20 upregulated and 38 downregulated metabolites (Figure 4c). The details of all identified fecal differential metabolites are listed in Appendix A. The heatmap of differential metabolites in the SM and SM-DMY groups was plotted to demonstrate the hierarchical clustering of altered metabolites in different groups (Figure 4d). The fecal differential metabolites in SM cows induced by DMY intervention were mainly enriched in pathways including alanine, aspartate, and glutamate metabolism; vitamin B6 metabolism, cortisol synthesis and secretion; protein digestion and absorption; aminoacyl-tRNA biosynthesis; amino acid biosynthesis; the mTOR signaling pathway; and steroid hormone biosynthesis (Figure 4e).

Machine learning analysis based on random forest algorithms was carried out to further characterize the fecal signature differential metabolites between different groups. As shown in Figure 5a, the importance of multiple differential metabolites such as (S)-indenestrol A, Tryptamine, Docosapentaenoic acid (22n-3), N-Acetyl-L-aspartic acid, L-Glutamine, Prostaglandin B1, Gamma-Tocotrienol, and 15-Deoxy-d-12,14-PGJ2 ranked higher within the random forest model. The abundance of (S)-indenestrol A (*p* < 0.05), Tryptamine (*p* < 0.001), N-Acetyl-L-aspartic acid (*p* < 0.05), Prostaglandin B1 (*p* < 0.05), Palmitoleic acid (*p* < 0.05), 15-Deoxy-d-12,14-PGJ2 (*p* < 0.05), and 6-Ketoprostaglandin E1 (*p* < 0.05) were remarkably decreased in the SM-DMY group in comparison with the SM group. Nevertheless, the abundance of Docosapentaenoic acid (22n-3) (*p* < 0.01), L-Glutamine (*p* < 0.05), Gamma-Tocotrienol (*p* < 0.05), L-Leucine (*p* < 0.05), L-Malic acid (*p* < 0.05), Allocholic acid (*p* < 0.05), L-Asparagine (*p* < 0.05), beta-Alanyl-L-arginine (*p* < 0.05), 8,11,14-Eicosatrienoic acid (*p* < 0.05), and L-Theanine (*p* < 0.05) were markedly heightened in the SM-DMY group compared with the SM group (Figure 5b). Receiver operating characteristic (ROC) analysis was then conducted to assess whether the differentially expressed metabolites were crucial for the differentiation between groups. The results showed that the area under the curve (AUC) values for 8,11,14-Eicosatrienoic acid, 15-Deoxy-d-12,14-PGJ2, 6-Ketoprostaglandin E1, Prostaglandin B1, Docosapentaenoic acid (22n-3), Allocholic acid, and L-Theanine were all above 0.80, indicating that these differential metabolites are representative and may contribute to elucidating the impact of DMY on the metabolic changes in intestinal flora (Figure 5c).

### 3.6. Effects of Dietary DMY Supplementation on Plasma Metabolomic Profiles

The blood metabolome serves as a critical tool for investigating the relationship between gut microbiota composition and host phenotypes. Additionally, we performed an analysis of the impact of DMY intervention on plasma metabolomic profiles of SM cows. We detected 6918 and 7817 metabolites in all plasma samples under positive and negative ion mode, respectively, among which a total of 6753 named metabolites were ultimately identified. The PLS-DA analysis of metabolite abundance of each plasma sample in the SM and SM-DMY groups showed that the contributions of PC1 and PC2 to the variation were 12.7% and 10.5% in positive ion mode and 11.4% and 11.7% in negative ion mode, respectively (Figure 6a,b). Cluster analysis of plasma differential metabolites presented obvious partitions, which suggested that there were significant differences in the composition of plasma metabolites between the SM group and SM-DMY group (Figure 6d). A volcano plot illustrating differential metabolites between groups demonstrated that a total of 420 differential metabolites, with 199 downregulated and 221 upregulated, were identified in the SM-DMY group compared with the SM group (Figure 6c). The details of plasma differential metabolites are presented in Appendix A. KEGG pathway enrichment analysis showed that the plasma differential metabolites in SM cows induced by DMY intervention were primarily enriched in pyrimidine metabolism, cysteine and methionine metabolism, glutathione metabolism, tryptophan metabolism, pantothenate and CoA biosynthesis, and inflammatory mediator regulation of TRP channels (Figure 6e).

The random forest plots of top 50 plasma differential metabolites in the SM vs. SM-DMY group, ranked by their importance as determined through machine learning analysis, are displayed in Figure 7a. The abundance of Dihydro-3-coumaricacid (*p* < 0.001), N-butyryl-L-Homoserinelactone-d5 (*p* < 0.01), Deoxycholic acid (*p* < 0.05), L-Ascorbic acid (*p* < 0.01), Theanine (*p* < 0.05), 5-Hydroxykynurenamine (*p* < 0.01), (R)-3-Amino-4-phenylbutyric acid (*p* < 0.05), Ethyl 4-(acetylthio)butyrate (*p* < 0.05), and D-(+)-Pantothenic acid (*p* < 0.01) were notably elevated in the SM-DMY group compared with the SM group. Conversely, the abundance of Stearoyl Serotonin (*p* < 0.001), Muramic acid (*p* < 0.01), 4-Methylhippuric acid (*p* < 0.05), Glycerol (*p* < 0.05), 20-Hydroxyeicosatetraenoic acid (*p* < 0.01), Docosatetraenoic acid (*p* < 0.05), ((+−))5-HETrE (*p* < 0.05), 8,20-DiHETE (*p* < 0.01), 16-Hydroxy-10-oxohexadecanoic acid (*p* < 0.05), 12,13-EODE (*p* < 0.05), and (9Z,12E)-15,16-dihydroxyoctadeca-9,12-dienoic acid (*p* < 0.05) were prominently decreased in the SM-DMY group relative to the SM group (Figure 7b). ROC analysis of plasma differential metabolites induced by DMY administration in SM cows revealed that the AUC values for Dihydro-3-coumaricacid, Stearoyl Serotonin, Deoxycholic acid, L-Ascorbic acid, Theanine, 5-Hydroxykynurenamine, Muramic acid, Docosatetraenoic acid, 20-Hydroxyeicosatetraenoic acid, and N-butyryl-L-Homoserinelactone-d5 were all above 0.80 (Figure 7c). These differential metabolites were typical to clarify the effects of DMY on body metabolic shift.

### 3.7. Associations Between Significantly Different Fecal Microbiome and Metabolites in Feces

Spearman’s rank correlation coefficient and significance test were applicated to analyze the correlations between gut differential microbes enriched by DMY intervention and fecal differential metabolites (Figure 8a). The level of 8,11,14-Eicosatrienoic acid was negatively correlated with the abundance of Cyanobacteria (r = −0.868, *p* = 0.0057) and *Phascolarctobacterium* (r = −0.829, *p* = 0.0025). Allocholic acid concentration showed a negative association with *Dehalobacterium* (r = −0.618, *p* = 0.019). For prostaglandins and their derivatives, 6-Ketoprostaglandin E1 level was negatively associated with *Coprococcus* (r = −0.749, *p* = 0.0020) and *Roseburia* (r = −0.648, *p* = 0.012); the abundance of Prostaglandin B1 was positively correlated with Elusimicrobia (r = 0.707, *p* = 0.0047) and Proteobacteria (r = 0.609, *p* = 0.021), but negatively correlated with *Roseburia* (r = −0.547, *p* = 0.043); the 15-Deoxy-D-12,14-PGJ2 level was positively associated with Proteobacteria (r = 0.688, *p* = 0.0065) while negatively correlated with Bacteroidetes (r = −0.600, *p* = 0.023). Among fatty acid metabolites, the abundance of Palmitoleic acid was positively correlated with Proteobacteria (r = 0.640, *p* = 0.014) and *Dehalobacterium* (r = 0.640, *p* = 0.014), but negatively correlated with *Roseburia* (r = −0.560, *p* = 0.037); Docosapentaenoic acid (22n-3) level was negatively associated with Cyanobacteria (r = −0.754, *p* = 0.0018), Elusimicrobia (r = −0.654, *p* = 0.011), Proteobacteria (r = −0.653, *p* = 0.011), *Dehalobacterium* (r = −0.591, *p* = 0.026), *Oscillospira* (r = −0.767, *p* = 0.0014), *Phascolarctobacterium* (r = −0.692, *p* = 0.0061), and *Akkermansia* (r = −0.776, *p* = 0.0011), while positively correlated with Bacteroidetes (r = 0.771, *p* = 0.0012), Actinobacteria (r = 0.613, *p* = 0.020), *Coprococcus* (r = 0.552, *p* = 0.041), and *Corynebacterium* (r = 0.749, *p* = 0.0020). With regard to the amino acid metabolites, L-Leucine showed negative associations with Elusimicrobia (r = −0.574, *p* = 0.032), Proteobacteria (r = −0.701, *p* = 0.0052), and *BF311* (r = −0.609, *p* = 0.021), while it exhibited positive associations with Actinobacteria (r = 0.692, *p* = 0.0061), *Coprococcus* (r = 0.653, *p* = 0.011), *Roseburia* (r = 0.710, *p* = 0.0045), *Corynebacterium* (r = 0.697, *p* = 0.0056) and *Corynebacterium variabile* (r = 0.726, *p* = 0.0033). L-Glutamine was negatively associated with Proteobacteria (r = −0.578, *p* = 0.030), *Dehalobacterium* (r = −0.569, *p* = 0.034), and *Oscillospira* (r = −0.807, *p* = 0.00049) but positively correlated with Bacteroidetes (r = 0.727, *p* = 0.0032) and *Corynebacterium variabile* (r = 0.691, *p* = 0.0062).

### 3.8. Correlation Analysis Among Significantly Different Fecal Microbiome, Plasma Metabolites, SCC, and Blood Parameters

The abundance of Cyanobacteria significantly downregulated by DMY intervention was positively correlated with milk SCC (r = 0.626, *p* = 0.017), plasma Hydroxycitric acid (r = 0.609, *p* = 0.021), and 20-Hydroxyeicosatetraenoic acid (r = 0.582, *p* = 0.029), while it was negatively correlated with the levels of serum T-AOC (r = −0.829, *p* = 0.00025), CAT (r = −0.585, *p* = 0.028), Theanine (r = −0.543, *p* = 0.045), D-(+)-pantothenic acid (r = −0.688, *p* = 0.0065), L-ascorbic acid (r = −0.631, *p* = 0.016), Dihydro-3-coumaric acid (r = −0.591, *p* = 0.026), and Equol 4-O-glucuronide (r = −0.569, *p* = 0.034). The abundance of Proteobacteria was positively associated with milk SCC (r = 0.538, *p* = 0.047) and the levels of MDA (r = 0.586, *p* = 0.028), Muramic acid (r = 0.538, *p* = 0.047), Stearoyl Serotonin (r = 0.653, *p* = 0.011), Kynurenic acid (r = 0.631, *p* = 0.016), Docosatetraenoic acid (r = 0.635, *p* = 0.015), 20-Hydroxyeicosatetraenoic acid (r = 0.802, *p* = 0.00056), and Glycerol (r = 0.736, *p* = 0.0027) but negatively associated with the levels of CAT (r = −0.697, *p* = 0.0056), Deoxycholic acid (r = −0.578, *p* = 0.030), Ethyl4-(acetylthio)butyrate (r = −0.653, *p* = 0.011), D-(+) -pantothenic acid (r = −0.653, *p* = 0.011), L-Ascorbic acid (r = −0.631, *p* = 0.016), Dihydro-3-coumaric acid (r = −0.723, *p* = 0.0035), Equol4-O-glucuronide (r = −0.723, *p* = 0.0035), and N-butyryl-L-Homoserine lactone-d5 (r = −0.727, *p* = 0.0032) in plasma. The abundance of *Dehalobacterium* presented positive correlations with milk SCC (r = 0.681, *p* = 0.0073), the levels of MDA (r = 0.611, *p* = 0.020), 9(Z),11(E)-Conjugated Linoleic Acid (r = 0.596, *p* = 0.025), Stearoyl Serotonin (r = 0.758, *p* = 0.0017), Kynurenic acid (r = 0.538, *p* = 0.047), Docosatetraenoic acid (r = 0.565, *p* = 0.035), 20-Hydroxyeicosatetraenoic acid (r = 0.604, *p* = 0.022), and Glycerol (r = 0.534, *p* = 0.049), while it exhibited negative associations with the levels of T-AOC (r = −0.644, *p* = 0.013), Theanine (r = −0.560, *p* = 0.037), 3-Methylhistamine (r = −0.704, *p* = 0.0049), Deoxycholic acid (r = −0.657, *p* = 0.011), (R)-3-Amino-4-phenylbutyric acid (r = −0.618, *p* = 0.019), Dihydro-3-coumaric acid (r = −0.653, *p* = 0.011), and Equol 4-O-glucuronide (r = −0.538, *p* = 0.047) in plasma. *BF311* was positively correlated with milk SCC (r = 0.577, *p* = 0.031), the levels of Stearoyl Serotonin (r = 0.596, *p* = 0.025), Docosatetraenoic acid (r = 0.635, *p* = 0.015), and 20-Hydroxyeicosatetraenoic acid (r = 0.565, *p* = 0.035) but negatively associated with the levels of T-AOC (r = −0.679, *p* = 0.0076), 3-Methylhistamine (r = −0.567, *p* = 0.034), Dihydro-3-coumaric acid (r = −0.587, *p* = 0.027), Equol4-O-glucuronide (r = −0.613, *p* = 0.020), and N-butyryl-L-Homoserine lactone-d5 (r = −0.679, *p* = 0.0076) in plasma (Figure 8b).

Furthermore, the abundance of *Coprococcus* significantly upregulated by DMY administration was negatively correlated with milk SCC (r = −0.535, *p* = 0.048), the levels of 9(Z),11(E)-Conjugated Linoleic Acid (r = −0.662, *p* = 0.010), and Stearoyl Serotonin (r = −0.604, *p* = 0.022) but positively associated with the levels of Deoxycholic acid (r = 0.723, *p* = 0.0035), Ethyl4-(acetylthio)butyrate (r = 0.613, *p* = 0.020), L-Ascorbic acid (r = 0.626, *p* = 0.017), Equol4-O-glucuronide (r = 0.543, *p* = 0.045), Niacinamide (r = 0.578, *p* = 0.030), and N-butyryl-L-Homoserine lactone-d5 (r = 0.666, *p* = 0.0093) in plasma. *Roseburia* showed negative correlations with milk SCC (r = −0.606, *p* = 0.022), plasma 9(Z),11(E)-Conjugated Linoleic Acid (r = −0.754, *p* = 0.0018), Stearoyl Serotonin (r = −0.776, *p* = 0.0011), and Glycerol (r = −0.556, *p* = 0.039), while it showed positive associations with the levels of 3-Methylhistamine (r = 0.629, *p* = 0.016), Deoxycholic acid (r = 0.811, *p* = 0.00043), L-Ascorbic acid (r = 0.556, *p* = 0.039), Dihydro-3-coumaric acid (r = 0.556, *p* = 0.039), Niacinamide (r = 0.560, *p* = 0.037), and N-butyryl-L-Homoserine lactone-d5 (r = 0.807, *p* = 0.00049) (Figure 8b).

## 4. Discussion

A previous study proved that cows with SM exhibit disordered gut microbiota and enhanced oxidative stress [14]. The present study explored the impacts of dietary supplementation with DMY on lactation performance, blood biochemical indexes, antioxidant capacity, and cytokine levels and illustrated the alleviating effect and potential regulatory mechanism of DMY on SM based on gut microbiota structure and metabolite profiles by applying multi-omics technologies.

The milk yield is the most intuitive index to measure the productivity and economic benefits of dairy cows, and milk fat percentage and milk protein percentage are important references for evaluating the nutritional commercial values of milk. The results in the current study showed that DMY had no significant effects on milk yield or the content of milk ingredients including protein, fat, and MUN in SM cows. Milk SCC is an essential and widely used indicator to assess the hygienic quality of milk and the diagnosis of mastitis in dairy cows [37]. The SM cows fed with DMY for 60 days exhibited lower milk SCC. These results indicated that DMY intervention did not significantly affect milk yield or quality; however, the reduction in SCC suggested that DMY might possess beneficial effects on ameliorating SM.

Blood biochemical parameters could serve as important indicators for reflecting the metabolic status of the body. DMY intervention did not significantly affect the biochemical parameters related to liver function, lipid metabolism, and protein metabolism in the serum of SM cows, implying that DMY had no negative influence on the basic metabolic status of the SM cows. Previous research showed that Moringa leaf flavonoids did not affect the physiological levels of common biochemical indicators such as TP, ALB, GLB, HDL, and LDH [38] in the blood of dairy cows, which was similar to our findings. DMY, a dihydroflavonol flavonoid compound, has been confirmed to possess obvious antioxidant activity both in vitro and in vivo [25,39]. The elevated oxidative stress levels during mastitis are related to the excessive generation of ROS in the process of inflammatory reactions [14,40]. In the present study, the serum T-AOC and CAT activities of cows in the SM-DMY group were significantly higher than those in the SM group, and the MDA content was observably lower than that in the SM group, indicating that DMY intervention could improve the antioxidant capacity of SM cows, which was comparable with the results from Guo et al. [25]. On the other hand, DMY administration had no significant effects on the content of cytokines in the serum of SM cows. Numerous studies have corroborated the close relationship between oxidative stress and inflammation. The increased ROS production under oxidative stress can activate the NF-κB signaling pathway, thereby promoting the occurrence of inflammatory diseases [41]. Therefore, we speculate that DMY might attenuate oxidative stress associated with SM.

In order to identify the gut microbiota regulated by DMY intervention and explore the potential mechanisms by which DMY-regulated gut microbiota affect mastitis in dairy cows, multi-omics techniques were devoted to analyze the impacts of DMY on the fecal microbiome, as well as the characteristics of fecal and plasma metabolomes, in SM cows. PCoA of beta diversity presented separated clustering between the SM group and SM-DMY group, indicating that DMY intervention changed the intestinal microbiota of SM cows. Additionally, DMY administration significantly downregulated the abundance of Cyanobacteria, Proteobacteria, Elusimicrobia, *Phascolarctobacterium*, *Dehalobacterium,* and *Oscillospira*, whereas it prominently upregulated the abundance of Bacteroidetes, *Roseburia*, *Coprococcus,* and *Corynebacterium* in the feces of SM cows. Moreover, correlation analysis demonstrated that Cyanobacteria, Proteobacteria, and *Dehalobacterium* were notably positively correlated with milk SCC, while *Roseburia* and *Coprococcus* were markedly negatively associated with milk SCC. Research from Wang et al. showed that Cyanobacteria is signally enriched in the rumen of CM cows [42]; however, the relationship between Cyanobacteria and mastitis remains unclear. *Dehalobacterium*, a Gram-positive obligate anaerobe, was found to be significantly reduced in the rumen of high-yielding dairy cows [43]. However, there are few reports on the functional role of Cyanobacteria and *Dehalobacterium*. *Bacteroides* was reported to be one of the most commonly altered microbes in the gut of animals administrated with flavonoid or polyphenol [44], which is consistent with our findings. *Coprococcus* and *Roseburia*, the major butyric acid-producing bacteria, form a complex symbiotic network with other microorganisms in the gut, which plays a vital role in regulating host nutrient metabolism and maintaining intestinal health [45]. The commensal *Roseburia ininalis* alleviates M-FMT (fecal microbiota transplantation from donor cows with mastitis)-induced mastitis through limiting bacterial translocation from the gut into the mammary glands and suppressing the secretion of inflammatory cytokines and blood–milk barrier damage by producing butyrate in mice [46]. Furthermore, the results of PICRUSt showed that DMY intervention significantly downregulated the abundance of genes related to amino acid degradation, antibiotic resistance, and O-antigen building blocks biosynthesis (E. coli) of gut microbiota in SM cows, further suggesting that DMY might exert beneficial effects on alleviating SM. Thus, we surmise that the decreased abundance of Cyanobacteria, Proteobacteria, and *Dehalobacterium* and the increased abundance of *Roseburia* and *Coprococcus* in the gut of SM cows induced by DMY administration is closely related to the alleviation of SM by DMY in cows (Figure 9).

With respect to the fecal metabolomic profiling, a significant decrease in the abundance of 16-Hydroxy hexadecanoic acid, 15-deoxy-D-12,14-PGJ2, Prostaglandin B1, and 6-Ketoprostaglandin E1 was observed in the SM-DMY group. Conversely, the abundance of 8,11,14-Eicosatrienoic acid, Allocholic acid, Docosapentaenoic acid (22n-3), Gamma-Tocotrienol, and L-Theanine was notably increased in the SM-DMY group. 8,11,14-Eicosatrienoic acid is a derivative of γ-linolenic acid that contributes to regulating the immune response and reducing the production of inflammatory mediators [47]. Allocholic acid, an SBA produced by intestinal microorganisms, not only participates in cholesterol metabolism but also regulates the composition of the gut microbiota, which plays a crucial role in maintaining intestinal health [48,49]. Correlation analysis showed that *Dehalobacterium* was significantly negatively associated with allocholic acid. Docosapentaenoic acid (22n-3) (DPA), a member of the ω-3 long-chain polyunsaturated fatty acid (PUFA) family known for its anti-inflammatory properties, is similar to eicosapentaenoic acid (EPA) and docosahexaenoic acid (DHA). DPA has been reported to reduce the production of pro-inflammatory substances such as prostaglandins and leukotrienes by inhibiting arachidonic acid metabolism [50]. Gamma-Tocotrienol, a member of the vitamin E family, exhibits potent antioxidant activity, which can effectively scavenge free radicals, ameliorate oxidative damage, and reduce the production of inflammatory mediators, thereby combating the occurrence of inflammatory diseases [51,52]. L-Theanine, systematically named N-ethyl-L-glutamine, has been proved to play an immunoregulatory role in inflammation, nerve damage, and intestinal immunity by activating γδT lymphocyte function, promoting glutathione (GSH) synthesis, and regulating the secretion of cytokines and neurotransmitters [53,54]. 15-deoxy-D-12,14-PGJ2, Prostaglandin B1, and 6-Ketoprostaglandin E1 belong to the prostaglandin family and its analogues, which are primarily synthesized from arachidonic acid via the cyclooxygenase (COX) pathway and subsequently metabolized in the body. These compounds play a critical physiological role in regulating inflammation, adipose tissue metabolism, and platelet function [55]. Correlation analysis indicated that Proteobacteria were positively correlated with 15-deoxy-D-12,14-PGJ2 and Prostaglandin B1, *Roseburia,* and *Coprococcus* were negatively associated with 6-Ketoprostaglandin E1. Therefore, the changes in the abundance of anti-inflammatory, antioxidant, and lipid metabolism-regulating molecules in the feces of SM cows induced by DMY intervention might play an important regulatory role in attenuating SM (Figure 9).

Metabolomic profiling of plasma revealed that the abundance of multiple antioxidant metabolites and SBAs was also notably upregulated in the SM-DMY group, including Deoxycholic acid, Theanine, Dihydro-3-coumaric acid, etc. Deoxycholic acid, as a common SBA, not only regulates MAPK and NF-κB signaling pathways to diminish the production of pro-inflammatory cytokines by activating the farnesoid X receptor (FXR) or G-protein coupled bile acid receptor 1 (GPBAR1), but also promotes the polarization of M2 macrophages and the proliferation of regulatory T cells (Tregs) to exert anti-inflammatory effects [56]. In addition, the abundance of butyric acid derivatives (Ethyl 4-(acetylthio)butyrate, (R)-3-Amino-4-phenylbutyric acid) and antioxidant compounds (Theanine, Dihydro-3-coumaric acid, L-Ascorbic acid, 5-Hydroxykynurenamine) in the plasma of cows in the SM-DMY group was significantly improved. Notably, the observed change in the abundance of Theanine in plasma was consistent with the result obtained from the fecal metabolome. Dihydro-3-coumaric acid is a phenolic acid compound, formed by the reduction of coumaric acid, which possesses antioxidant activity and can be used as a free radical scavenger. Rafiee et al. demonstrated that coumaric acid mitigates doxorubicin-induced nephrotoxicity through suppression of oxidative stress, inflammation, and apoptosis [57]. 5-Hydroxykynurenamine is an important metabolite in the kynurenine pathway of tryptophan metabolism, which modulates the secretion of pro-inflammatory cytokines by activating GPR35 and AhR signaling pathways [58,59]. Furthermore, N-butyryl-L-Homoserine lactone-d5, as an important quorum-sensing signaling molecule, was proved to exert antibacterial activity by suppressing the biofilm formation in *Pseudomonas aeruginosa* [60]. 3-Methylhistamine is an end product of histamine metabolism. DMY intervention markedly increased the abundance of 3-Methylhistamine in the plasma of SM cows, implying that DMY might promote the degradation of histamine, thereby potentially mitigating inflammation in dairy cows.

Meanwhile, the plasma levels of pro-inflammatory molecules were markedly reduced by DMY intervention in SM cows, including eicosanoid metabolites, ω-6-PUFA, and structural components of bacteria. 20-Hydroxyeicosatetraenoic acid (20-HETE) is one of the principal cytochrome P450-derived vasoactive eicosanoids, and its abnormal metabolism is closely related to the occurrence and progression of inflammation, hypertension, and cardiovascular diseases [61,62]. 8,20-DiHETE is a metabolite formed by hydroxylation of the eighth carbon atom of 20-HETE. Docosatetraenoic acid (DTA), belonging to ω-6 long-chain PUFAs, can be metabolized into bioactive compounds, such as prostaglandins and leukotrienes, which are crucial in mediating inflammatory response [63]. ((+-))5-HETrE (5-Hydroxyeicosatrienoic acid), a metabolite of ω-6 PUFA γ-linolenic acid, has been proved to be significantly elevated in the serum of obese mice [64]. Correlation analysis showed that both 20-Hydroxyeicosatetraenoic acid and Docosatetraenoic acid, which were significantly downregulated in the plasma of SM cows fed with DMY in the current study, presented positive linkages with Proteobacteria and *Dehalobacterium*. Stearoyl Serotonin is a derivative of 5-hydroxytryptamine, with few studies on its physiological function. However, recent research indicates that 5-hydroxytryptamine affects the release of inflammatory cytokines by interacting with serotonin receptors located on the surface of immune cells, as well as influencing the balance of Th1/Th2 cytokines, thereby participating in the regulation of immune function [65]. Muramic acid is a major component of peptidoglycan in bacterial cell walls. As an exclusive molecule of bacteria, Muramic acid is one of the decisive markers for the immune system of the host to recognize bacteria and trigger an immune response [66]. In this study, Muramic acid showed a positive correlation with Proteobacteria, and the abundance of Muramic acid was prominently diminished in the plasma of SM cows administered DMY. It is hypothesized that DMY intervention might ameliorate mastitis via improving intestinal barrier function and preventing harmful gut microbes and their pro-inflammatory metabolites from entering the circulatory system.

Intriguingly, the relative abundance of multiple gut microbes and metabolites previously identified to be significantly upregulated or downregulated in SM cows [67] showed completely opposite changes in the SM-DMY group of the present study, including *Proteobacteria*, *Cyanobacteria*, *YS2*, *RFP12*, *Coprococcus*, Stearoyl Serotonin, Deoxycholic acid, Ethyl 4-(acetylthio)butyrate, Glycerol, Prostaglandin B1, Gamma-Tocotrienol, etc. These shifts in the gut microbial community structure and metabolite profiles, characterized by a significant increase in beneficial genera and antioxidant metabolites as well as a concurrent decrease in potentially detrimental taxa and pro-inflammatory mediators, support the concept that DMY may act not only as a broad-spectrum antioxidant but also as a modulator of host–microbiota metabolism. Therefore, DMY undoubtedly contributes to the observed systemic improvements in oxidative stress of SM cows; the multi-omics data of the current study further suggested that its beneficial effects might extend beyond non-specific radical scavenging. However, the specific molecular mechanisms by which DMY regulated the key differential microbes or metabolites identified in this study to alleviate mastitis in dairy cows remains unclear and need to be further elucidated in future studies.

## 5. Conclusions

In conclusion, this study probed into the beneficial effects and potential regulatory mechanisms of DMY in dairy cows with SM. The results revealed that dietary supplementation with DMY could mitigate SM through modulating the intestinal microbial community, considerably reducing the prevalence of potentially pathogenic bacteria, increasing the abundance of beneficial bacteria, elevating the levels of anti-inflammatory and antioxidant metabolites, and strengthening the overall antioxidant capacity of dairy cows. The current study provided a new perspective for the development of DMY as a phytogenic dietary supplement in modulating host–microbiota interactions to combat inflammatory diseases of the udder in dairy cows.

## Figures and Tables

**Figure 1 microorganisms-13-01890-f001:**
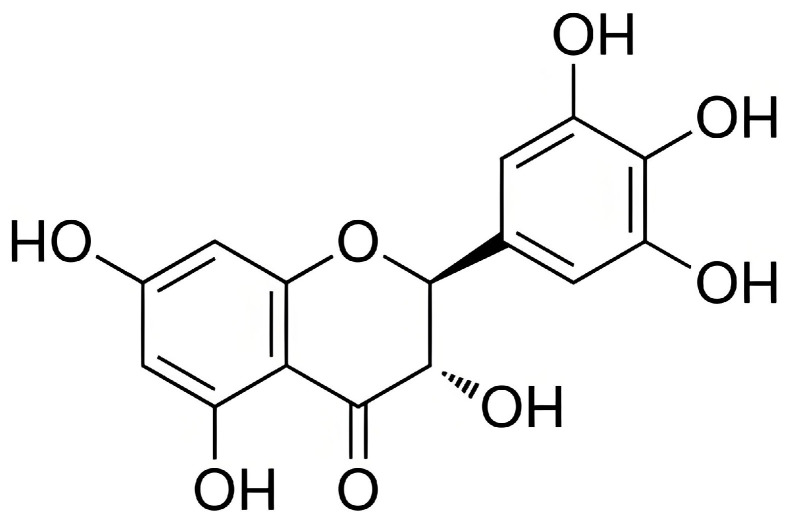
The chemical structure of DMY (Adapted with permission from ref. [15]. Copyright © 2002 Elsevier Science B.V.).

**Figure 2 microorganisms-13-01890-f002:**
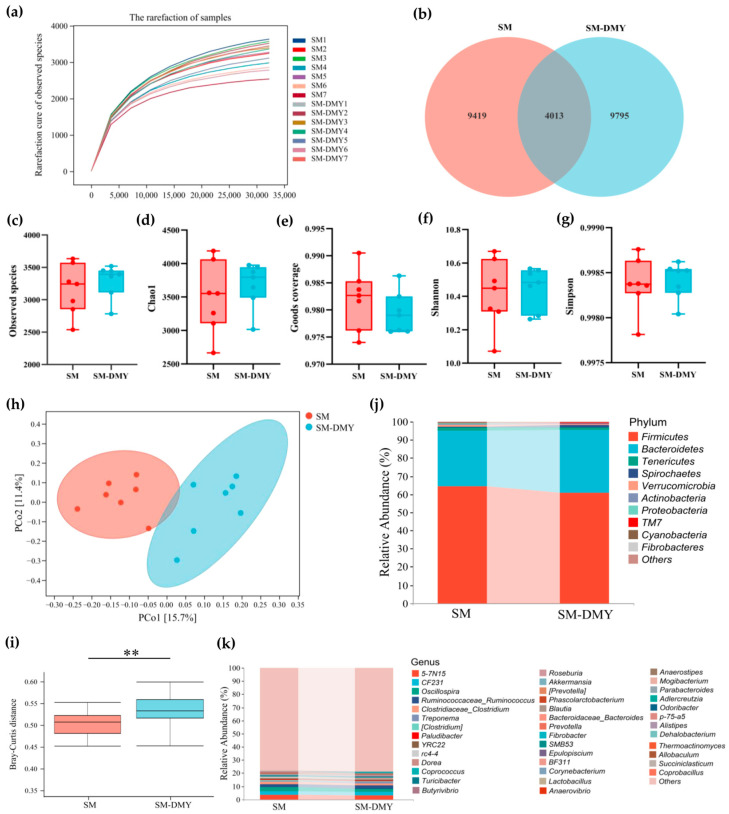
Effects of dietary supplementation with DMY on the gut microbial diversity of cows with SM. (**a**) The rarefaction curves of observed species for all samples; (**b**) Venn diagram of ASV distribution between groups; (**c**–**g**) boxplots of α diversity indexes: observed species, Chao1, Goods coverage, Shannon and Simpson, respectively; (**h**) β diversity between groups analyzed by PCoA; (**i**) Bray–Curtis distance between groups; (**j**,**k**) the column diagram of taxonomic composition analysis of bacteria at phylum and genus level, respectively. The results are presented as mean ± SEM (*n* = 7). ** *p* < 0.01.

**Figure 3 microorganisms-13-01890-f003:**
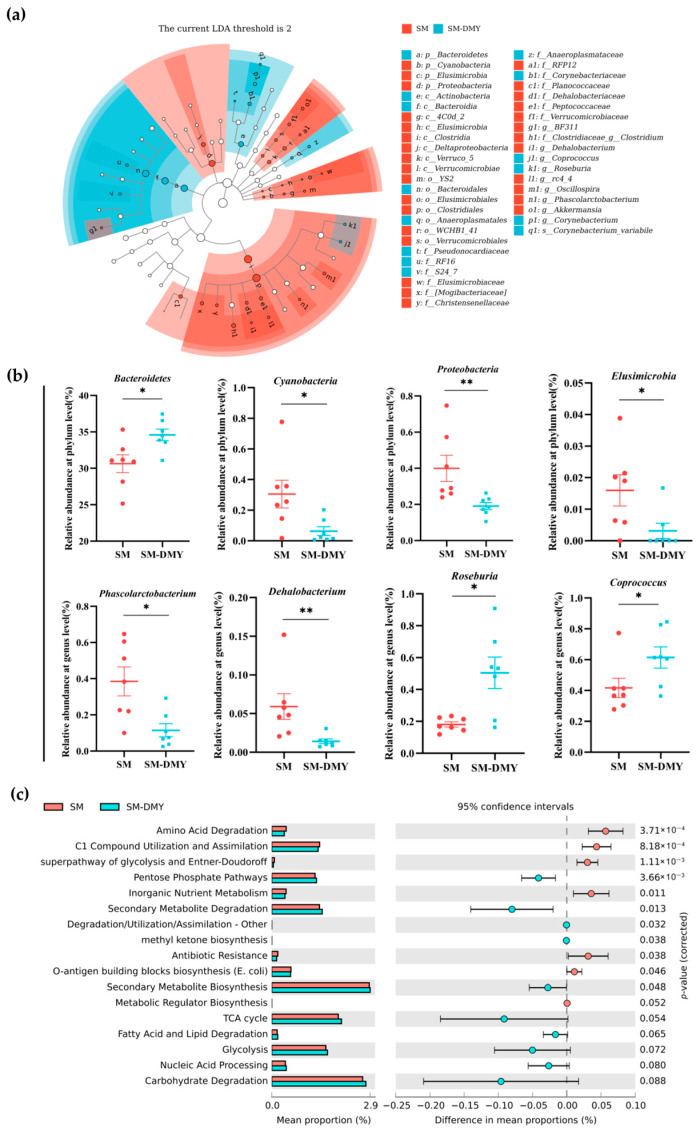
Effects of dietary supplementation with DMY on the gut microbial composition and function in cows with SM. (**a**) LEfSe analysis of differential intestinal microbes between the SM group and SM-DMY group; (**b**) scatter plot of the relative abundance of differential intestinal bacteria between the SM group and SM-DMY group at several taxonomic levels. Data are presented as mean ± SEM (*n* = 7), * *p* < 0.05, ** *p* < 0.01. (**c**) PICRUSt analysis of the differences in gut microbial function between the SM group and SM-DMY group. The abundance proportion of different functional pathways of gut microbiota with a corrected *p*-value < 0.1 are presented. The central plot denotes the difference in proportion of functional abundance within a 95% confidence interval.

**Figure 4 microorganisms-13-01890-f004:**
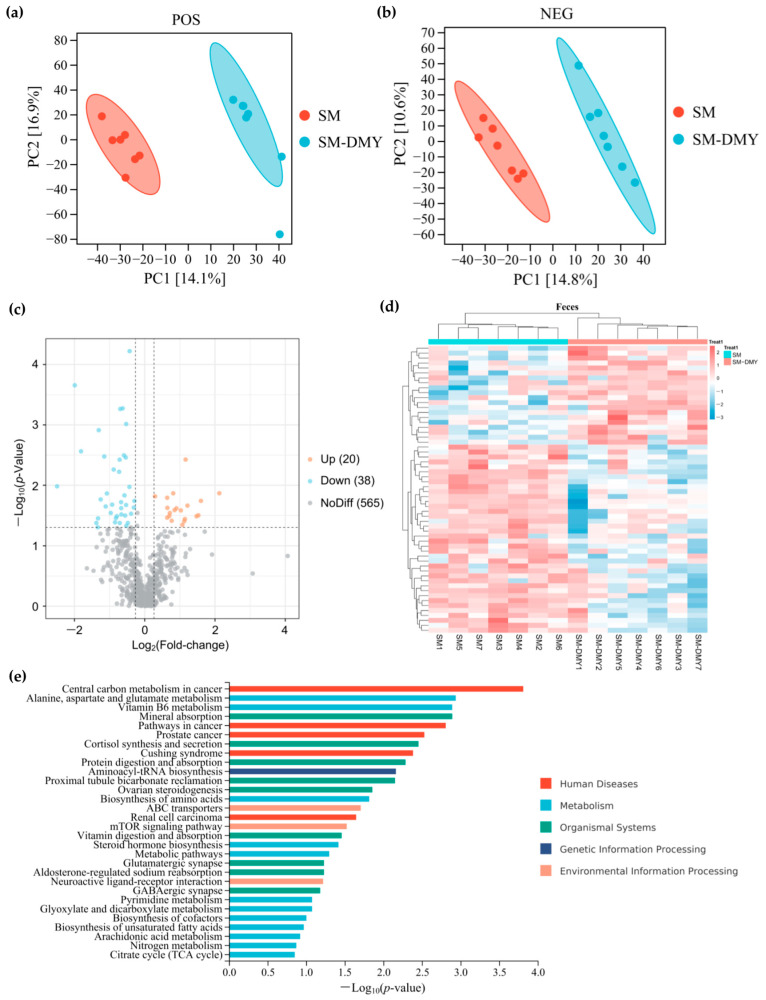
Effects of dietary supplementation with DMY on the fecal metabolite profiles in cows with SM. (**a**) PLS-DA analysis of metabolites in each sample under positive ion mode; (**b**) PLS-DA analysis of metabolites in each sample under negative ion mode; (**c**) volcano plot of fecal differential metabolites; (**d**) clustering heatmap of fecal differential metabolites; (**e**) KEGG enrichment analysis of fecal differential metabolites between the SM group and SM-DMY group.

**Figure 5 microorganisms-13-01890-f005:**
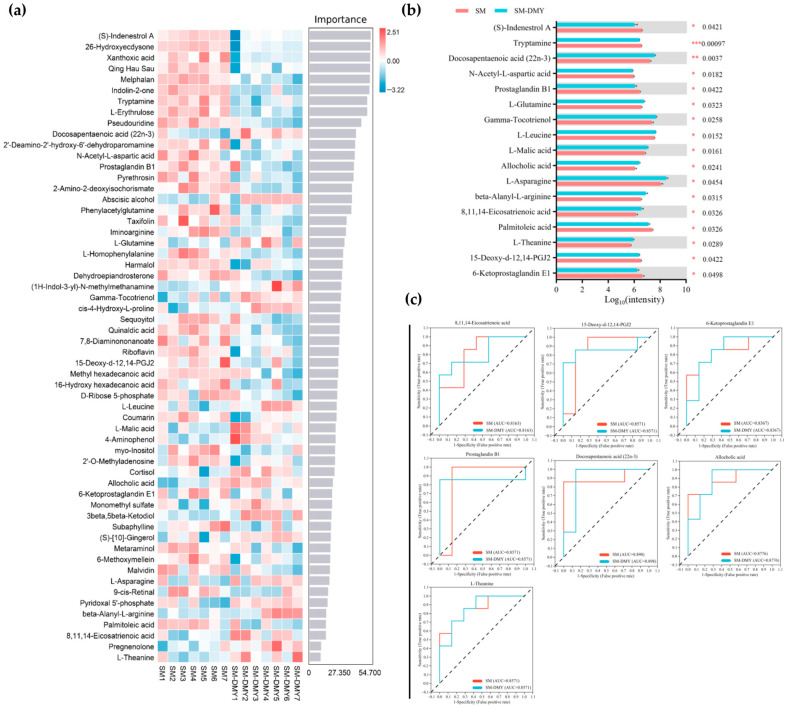
Effects of DMY intervention on the fecal signature differential metabolites in cows with SM. (**a**) Machine-learning analysis of fecal differential metabolites between the SM group and SM-DMY group. (**b**) Significant differential metabolites in feces samples among the SM group and SM-DMY group. Data are presented as mean ± SEM (*n* = 7), * *p* < 0.05, ** *p* < 0.01, *** *p* < 0.001. (**c**) Receiver operating characteristic (ROC) curves to estimate the significant fecal differential metabolites, which are critical for intergroup differentiation. The dotted line represents the performance benchmark of a completely random classifier. The farther the ROC curve is from the benchmark line, the better the predictive performance of the model.

**Figure 6 microorganisms-13-01890-f006:**
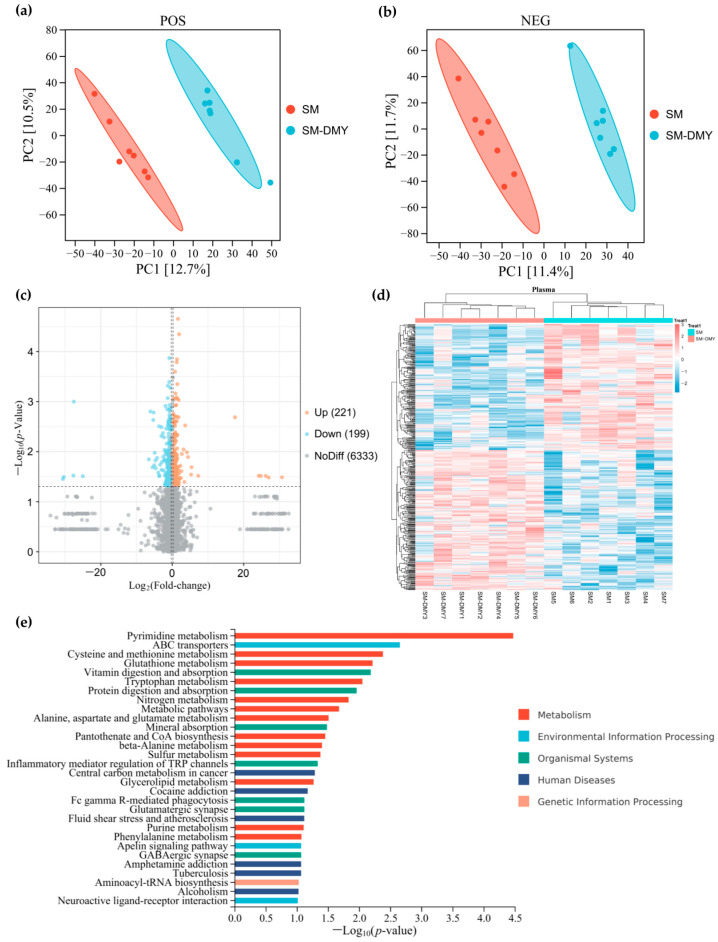
Effects of dietary supplementation with DMY on the plasma differential metabolites in cows with SM. (**a**) PLS-DA analysis of metabolites in each sample under positive ion mode; (**b**) PLS-DA analysis of metabolites in each sample under negative ion mode; (**c**) volcano plot of plasma differential metabolites; (**d**) clustering heatmap of plasma differential metabolites; (**e**) KEGG enrichment analysis of plasma differential metabolites between the SM group and SM-DMY group.

**Figure 7 microorganisms-13-01890-f007:**
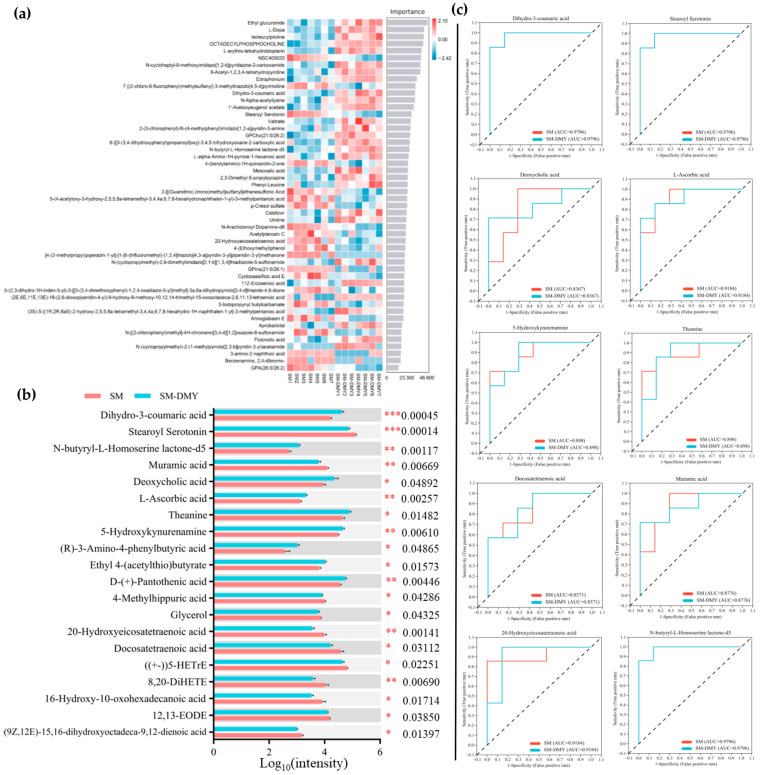
Effects of DMY intervention on the plasma signature differential metabolites in cows with SM. (**a**) Machine-learning analysis of plasma differential metabolites between the SM group and SM-DMY group. (**b**) Significant differential metabolites in plasma samples between the SM group and SM-DMY group. Data are presented as mean ± SEM (*n* = 7), * *p* < 0.05, ** *p* < 0.01, *** *p* < 0.001. (**c**) Receiver operating characteristic (ROC) curves to estimate the significant plasma differential metabolites, which are critical for intergroup differentiation. The dotted line represents the performance benchmark of a completely random classifier. The farther the ROC curve is from the benchmark line, the better the predictive performance of the model.

**Figure 8 microorganisms-13-01890-f008:**
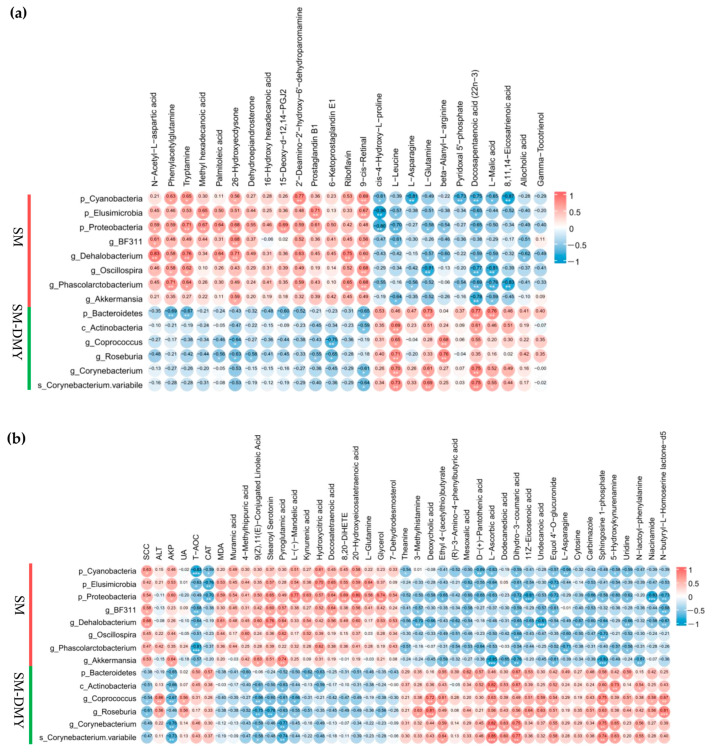
Spearman correlation analysis of (**a**) gut microbiome and fecal metabolome, (**b**) gut microbiome, SCC, blood parameters, and plasma metabolome between the SM and SM-DMY groups. The legend illustrates the values of correlation coefficients. Positive correlation is indicated by the red color, whereas negative correlation is represented by blue. The strength of the correlation is reflected by the intensity of the colors. The significance is displayed as * *p* < 0.05, ** *p* < 0.01, *** *p* < 0.001.

**Figure 9 microorganisms-13-01890-f009:**
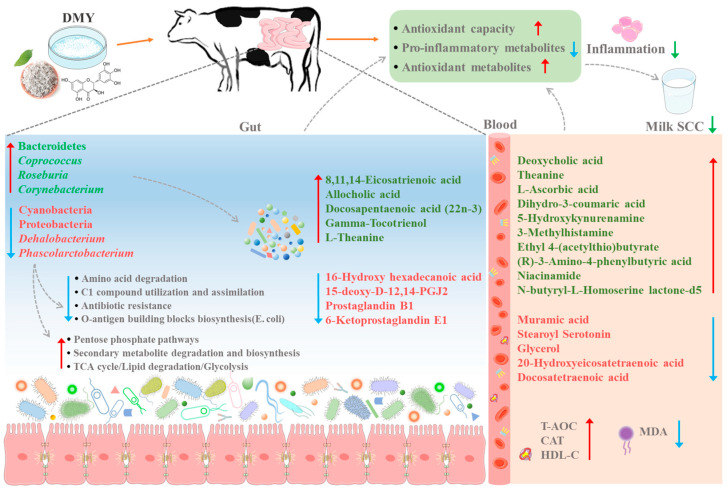
A schematic diagram to illustrate the potential mechanism by which DMY alleviates SM through modulating hindgut microbiota and metabolism in dairy cows. DMY, dihydromyricetin; SCC, somatic cell count; T-AOC, total antioxidant capacity; CAT, catalase; MDA, malondialdehyde; HDL-C, high-density lipoprotein cholesterol. The upward red solid arrows and downward blue solid arrows indicate upregulated or downregulated microbes, metabolites, indexes, and functions, respectively.

**Table 1 microorganisms-13-01890-t001:** Effects of dietary supplementation with DMY on milk yield and compositions of dairy cows with SM.

Items	Time	SM	SM-DMY	*p*-Value
Milk yield (kg/d)	0 d	19.48 ± 1.98	19.53 ± 1.99	0.9860
	60 d	13.73 ± 2.83	15.87 ± 1.17	0.4283
Milk fat (%)	0 d	4.28 ± 0.38	3.89 ± 0.29	0.4225
	60 d	3.62 ± 0.55	3.81 ± 0.21	0.7090
Milk protein (%)	0 d	3.30 ± 0.10	3.28 ± 0.078	0.8978
	60 d	3.44 ± 0.074	3.34 ± 0.055	0.2979
F/P	0 d	1.30 ± 0.11	1.20 ± 0.11	0.5075
	60 d	1.05 ± 0.14	1.14 ± 0.057	0.4895
SCC (×10^4^ cells/mL)	0 d	64 ± 9.0	64 ± 9.7	0.9873
	60 d	183 ± 113.0 ^a^	28 ± 13.4 ^b^	0.0064
MUN (mg/dL)	0 d	16.27 ± 0.31	16.32 ± 0.37	0.9187
	60 d	19.00 ± 0.80	19.60 ± 0.73	0.6112

^a, b^ Values within a row with different letters differed significantly (*p* < 0.05). Data are presented as mean ± SEM. Abbreviations: SCC, somatic cell count; MUN, milk urea nitrogen.

**Table 2 microorganisms-13-01890-t002:** Effects of dietary supplementation with DMY on blood biochemical indices of dairy cows with SM.

Items	SM	SM-DMY	*p*-Value
TP (g/L)	71.34 ± 1.50	74.61 ± 1.39	0.1357
ALB (g/L)	32.79 ± 0.85	33.50 ± 0.56	0.4976
GLB (g/L)	38.56 ± 1.94	41.11 ± 1.63	0.3336
AST (U/L)	73.96 ± 8.54	69.07 ± 3.23	0.6024
ALT (U/L)	26.24 ± 1.86	28.49 ± 1.83	0.4064
AKP (U/L)	355.66 ± 38.94	273.15 ± 20.74	0.0860
BUN (mmol/L)	6.32 ± 0.38	6.11 ± 0.32	0.6857
UA (µmol/L)	30.60 ± 3.55	35.91 ± 2.45	0.2411
TG (mmol/L)	0.18 ± 0.035	0.12 ± 0.0088	0.1405
TC (mmol/L)	2.49 ± 0.23	3.03 ± 0.29	0.1744
HDL-C (mmol/L)	1.66 ± 0.15	2.10 ± 0.16	0.0656
LDL-C (mmol/L)	0.59 ± 0.063	0.72 ± 0.12	0.3504
NEFA (mmol/L)	0.12 ± 0.023	0.095 ± 0.014	0.4348
LDH (U/L)	1440.92 ± 66.44	1444.62 ± 39.72	0.9626

Data are presented as mean ± SEM. Abbreviations: TP, total protein; ALB, albumin; GLB, globulin; AST, aspartate transaminase; ALT, alanine aminotransferase; AKP, alkaline phosphatase; BUN, blood urea nitrogen; UA, uric acid; TG, triglyceride; TC, total cholesterol; HDL-C, high-density lipoprotein cholesterol; LDL-C, low-density lipoprotein cholesterol; NEFA, non-esterified fatty acid; LDH, lactate dehydrogenase.

**Table 3 microorganisms-13-01890-t003:** Effects of dietary supplementation with DMY on cytokine levels in the serum of dairy cows with SM.

Items	SM	SM-DMY	*p*-Value
IL-1β (ng/L)	423.91 ± 36.75	451.92 ± 12.44	0.4842
IL-2 (ng/L)	265.40 ± 24.31	283.16 ± 11.58	0.5221
TNF-α (ng/L)	165.74 ± 11.60	165.72 ± 4.81	0.9988
IL-8 (ng/L)	161.48 ± 9.79	175.02 ± 4.77	0.2489
IL-10 (ng/L)	122.13 ± 8.51	129.82 ± 8.98	0.5458

Data are presented as mean ± SEM. Abbreviations: TNF-α, tumor necrosis factor α; IL-1β, interleukin 1β; IL-2, interleukin 2; IL-8, interleukin 8; IL-10, interleukin 10.

**Table 4 microorganisms-13-01890-t004:** Effects of dietary supplementation with DMY on antioxidant capacity of dairy cows with SM.

Items	SM	SM-DMY	*p*-Value
T-AOC (mM)	0.68 ± 0.026 ^a^	0.85 ± 0.039 ^b^	0.0035
SOD (U/mL)	13.46 ± 0.79	14.41 ± 0.48	0.3206
GSH-PX (U/mL)	239.99 ± 32.82	261.06 ± 12.66	0.5604
CAT (U/mL)	1.99 ± 0.41 ^a^	3.69 ± 0.65 ^b^	0.0462
MDA (nmol/mL)	3.11 ± 0.82 ^a^	1.45 ± 0.28 ^b^	0.0482

^a, b^ Values within a row with different letters differed significantly (*p* < 0.05). Data are presented as mean ± SEM. Abbreviations: T-AOC, total antioxidant capacity; SOD, superoxide dismutase; GSH-PX, glutathione peroxidase; CAT, catalase; MDA, malondialdehyde.

## Data Availability

The original sequencing data presented in the study are openly available in NCBI SRA database at https://www.ncbi.nlm.nih.gov/bioproject/, accessed on 13 July 2025, accession number: PRJNA1206781.

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
