# Peer review of "Investigating the Alleviating Effects of Dihydromyricetin on Subclinical Mastitis in Dairy Cows: Insights from Gut Microbiota and Metabolomic Analysis"

_microorganisms, 2025, doi:10.3390/microorganisms13081890_

Round 1
Reviewer 1 Report
Comments and Suggestions for Authors
- Consider reducing the number of figures and avoiding consecutive figure presentations, as this may hinder reader comprehension.
- It is recomendad to present the Spearman correlation in numerical form on page 17, line 468.
- In the discussion section, there appears to be an excessive reliance on citation number 25, which refers to work conducted by the same group of researchers.
Reviewer 2 Report
Comments and Suggestions for Authors
The paper describes a carefully planned and comprehensive investigation that looks into whether dihydromyricetin (DMY) could be used as a phytogenic product to help dairy cows with subclinical mastitis (SM). The multi-omics method that combines gut microbiota profiling and metabolomic analysis is a good idea that gives us new information about how the host, microbiota, and metabolites interact with each other.
The paper was submitted to the proper section of Microorganisms and is of interest to the readers. However, I have also some specific comments and suggestions for the authors on how to improve their work.
Detailed comments:
Abstract is significantly too long. Right now it is almost 400 words with the limit being 200. It must be shortened. I recommend removing some background information (Lines 16-26). Also, you don’t have to state (in abstract) the names of the compounds, i.e. lines 35-37. The readers would look into discussion anyway, and this is only making the abstract less useful and clear.
Line 58, what kind of antibiotics? Please be more specific.
Line 79, the structure of DMY should be presented in the introduction in a form of a figure
Lines 79–85: Summarize the structural and physicochemical characteristics of DMY that facilitate its intestinal or systemic bioactivity (e.g., solubility, stability, bioavailability), particularly in ruminants, where rumen metabolism may influence flavonoid disposition.
Line 82, other important sources of DMY are Hesperis pendula (DOI: 10.56782/pps.135 ) and Astragalus angustifolius (10.56782/pps.214), which should be referenced.
Lines 128-130, what kind of in vitro trial? Please be more specific and provided more details. The rationale for selecting 0.05% DMY needs clearer justification.
Table 1, SCC, those values are not rounded properly, the uncertainty should have at most 2 significant digits, so, i.e. it should be 180 +/- 120 (for SM)
Table 2, AKP, it should be 0.0860
In the discussion the authors should state whether the observed beneficial effects of DMY are solely due to (non specific) antioxidant properties or, maybe, are also associated with the specific mechanism of action. Maybe the DMY acts as an inhibitor of certain bacterial protein, i.e.?
At some point in the end of the discussion the authors should briefly state the limitations of current study.
Minor points:
Page 3, there’s a lot of empty space here, but maybe it is solely in the pdf version
Tables S1-S5 should be presented in one file
Round 2
Reviewer 2 Report
Comments and Suggestions for Authors
The Authors have revised and improved their work. Current version can be accepted as it is now.